# Learning Temporal Point Processes via Reinforcement Learning

**Shuang Li**[*1], **Shuai Xiao** [2], **Shixiang Zhu**[1], **Nan Du**[3], **Yao Xie**[1], and **Le Song**[1,2]

[1]Georgia Institute of Technology
[2]Ant Financial
[3]Google Brain

## Abstract

Social goods, such as healthcare, smart city, and information networks, often produce ordered event data in continuous time. The generative processes of these event data can be very complex, requiring flexible models to capture their dynamics. Temporal point processes offer an elegant framework for modeling event data without discretizing the time. However, the existing maximum-likelihood-estimation (MLE) learning paradigm requires hand-crafting the intensity function beforehand and cannot directly monitor the goodness-of-fit of the estimated model in the process of training. To alleviate the risk of model-misspecification in MLE, we propose to generate samples from the generative model and monitor the quality of the samples in the process of training until the samples and the real data are indistinguishable. We take inspiration from reinforcement learning (RL) and treat the generation of each event as the action taken by a stochastic policy. We parameterize the policy as a flexible recurrent neural network and gradually improve the policy to mimic the observed event distribution. Since the reward function is unknown in this setting, we uncover an analytic and nonparametric form of the reward function using an inverse reinforcement learning formulation. This new RL framework allows us to derive an efficient policy gradient algorithm for learning flexible point process models, and we show that it performs well in both synthetic and real data.

## 1 Introduction

Many natural and artificial systems produce a large volume of discrete events occurring in continuous time, for example, the occurrence of crime events, earthquakes, patient visits to hospitals, financial transactions, and user behavior in mobile applications [5]. It is essential to understand and model these complex and intricate event dynamics so that accurate prediction, recommendation or intervention can be carried out subsequently depending on the context.

Temporal point processes offer an elegant mathematical framework for modeling the generative processes of these event data. Typically, parametric (or semi-parametric) assumptions are made on the intensity function [11, 9] based on prior knowledge of the processes, and the maximum-likelihood-estimation (MLE) is used to fit the model parameters from data. These models often work well when the parametric assumptions are correct. However, in many cases where the real event generative process is unknown, these parametric assumptions may be too restricted and do not reflect the reality.

Thus there emerge some recent efforts in increasing the expressiveness of the intensity function using nonparametric forms [7] and recurrent neural networks [6, 19]. However, these more sophisticated models still rely on maximizing the likelihood which now involves intractable integrals and needs to be approximated. Most recently, [27] proposed to bypass the problem of maximum likelihood by adopting a generative adversarial network (GAN) framework, where a recurrent neural network is

---

[*]Correspondence to: Shuang Li <sli370@gatech.edu>, Yao Xie <yao.xie@isye.gatech.edu>, and Le Song <lsong@cc.gatech.edu>

learned to transform event sequence from a Poisson process to the target event sequence. However, this approach is rather computationally intensive, since it requires fitting another recurrent neural network as the discriminator, and it takes many iterations and careful tuning for both neural networks to reach equilibrium.

In this paper, we take a new perspective and establish an under-explored connection between temporal point processes and reinforcement learning: the generation of each event can be treated as the action taken by a stochastic policy, and the intensity function learning problem in temporal point processes can be viewed as the policy learning problem in reinforcement learning.

More specifically, we parameterize a stochastic policy $\pi$ using a recurrent neural network over event history and learn the unknown reward function via *inverse reinforcement learning* [1, 20, 28, 15]. Our algorithm for policy optimization iterates between learning the reward function and the stochastic policy $\pi$. Inverse reinforcement learning is known to be time-consuming, which requires solving a reinforcement learning problem in every inner-loop. To tackle this problem, we convert the inverse reinforcement learning step to a minimization problem over the discrepancy between the expert point process and the learner point process. By choosing the function class of reward to be the unit ball in reproducing kernel Hilbert space (RKHS) [13, 16, 8], we can get an explicit nonparametric closed form for the optimal reward function. Then the stochastic policy can be learned by a customized policy gradient with the optimal reward function having an analytical expression. An illustration of our modeling framework is shown in Figure 1.

We conducted experiments on various synthetic and real sequences of event data and showed that our approach outperforms the state-of-the-art regarding both data description and computational efficiency.

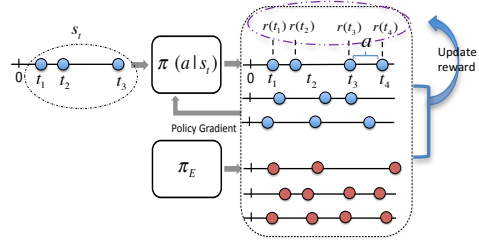

Figure 1: Illustration of our modeling framework. The observed trajectories of events will be viewed as the actions generated by an *expert* policy $\pi_E$. The goal is to learn a policy which we call *learner* that mimics the distribution of the observed expert event sequences. The learner policy $\pi(a|s_t)$ provides the probability of the next event occurring at $a$ after $t$, and $s_t := \{t_i\}_{t_i < t}$ is the history of events before $t$. We parametrize $\pi(a|s_t)$ by a recurrent neural network (RNN) with stochastic neurons [4], where the generated events are fed back to the RNN leading to a double stochastic point process [12]. Furthermore, each generated event $t_i$ will be also associated with a reward $r(t_i)$, and the policy will be learned by maximizing the expected cumulative rewards [26].

## 2  Preliminaries

**Temporal Point Processes.** A temporal point process is a stochastic process whose realization is a sequence of discrete events $\{t_i\}$ with $t_i \in \mathbb{R}^+$ and $i \in \mathbb{Z}^+$ abstracted as points on a timeline [5]. Let the history $s_t = \{t_1, t_2, \ldots, t_n | t_n < t\}$ be the sequence of event times up to but not including time $t$. The intensity function (rate function) $\lambda(t|s_t)$ conditioned on the history $s_t$ uniquely characterizes the generative process of the events. Different functional forms of $\lambda(t|s_t)dt$ capture different generating patterns of events. For example, a plain homogeneous Poisson process has $\lambda(t|s_t) = \lambda_0 \geqslant 0$, implying that each event occurs independently of each and uniformly on the timeline. A Hawkes process has $\lambda(t|s_t) = \lambda_0 + \sum_{t_i \in s_t} \exp(-(t - t_i))$ where the occurrences of past events will boost future occurrences. Given the intensity function, the survival function defined as $S(t|s_t) = \exp(-\int_{t_n}^{t} \lambda(\tau)d\tau)$ is the conditional probability that no event occurs in the window $[t_n, t)$, and the likelihood of observing event at time $t$ is defined as $f(t|s_t) = \lambda(t|s_t)S(t|s_t)$. Then we can express the joint likelihood of observing a sequence of events $s_T = \{t_1, t_2, \ldots, t_n | t_n < T\}$ up to an observation window $T$ as

$$p(\{t_1, t_2, \ldots, t_n | t_n < T\}) = \prod_{t_i \in s_T} \lambda(t_i | s_{t_i}) \cdot \exp\left(-\int_0^T \lambda(\tau | s_\tau)d\tau\right). \qquad (1)$$

The integral normalization in the likelihood function can be intensive to compute especially in cases where $\lambda(t|s_t)$ do not have a simple form. In this case, a numerical approximation is typically needed which may affect the accuracy of the fitting process.

**Reproducing Kernel Hilbert Spaces.** A reproducing kernel Hilbert space (RKHS) $\mathcal{H}$ on $\mathcal{T}$ with a kernel $k(t, t')$ is a Hilbert space of functions $f(\cdot) : \mathcal{T} \mapsto \mathbb{R}$ with inner product $\langle \cdot, \cdot \rangle_{\mathcal{H}}$. Its element $k(t, \cdot)$ satisfies the reproducing property: $\langle f(\cdot), k(t, \cdot) \rangle_{\mathcal{H}} = f(t)$, and consequently, $\langle k(t, \cdot), k(t', \cdot) \rangle_{\mathcal{H}} = k(t, t')$, meaning that we can view the evaluation of a function $f$ at any point

$t \in \mathcal{T}$ as an inner product. Commonly used RKHS kernel function includes Gaussian radial basis function (RBF) kernel $k(t, t') = \exp(- \|t - t'\|^2 / 2\sigma^2)$ where $\sigma > 0$ is the kernel bandwidth, and polynomial kernel $k(t, t') = (\langle t, t' \rangle + a)^d$ where $a > 0$ and $d \in \mathbb{N}$ [23, 3, 13]. In this paper, if not otherwise stated, we will assume that Gaussian RBF kernel is used. Let $\mathbb{P}$ be a measure on $\mathcal{T}$, we define the mapping of $\mathbb{P}$ to RKHS, $\mu_{\mathbb{P}} := \mathbb{E}_{\mathbb{P}}[k(t, \cdot)] = \int_{t \in \mathcal{T}} k(t, \cdot) \, d\mathbb{P}(t)$, as the Hilbert space embedding of $\mathbb{P}$ [24]. Then for all $f \in \mathcal{H}$, $\mathbb{E}_{\mathbb{P}}[f(t)] = \langle f, \mu_{\mathbb{P}} \rangle_{\mathcal{H}}$ by the reproducing property. Similarly, one can also embed another measure $\mathbb{Q}$ on $\mathcal{T}$ into RKHS as $\mu_{\mathbb{Q}}$. Then a distance between measure $\mathbb{P}$ and $\mathbb{Q}$ can be defined as $\|\mu_{\mathbb{P}} - \mu_{\mathbb{Q}}\|_{\mathcal{H}} := \sup_{\|f\|_{\mathcal{H}} \leqslant 1} \langle f, \mu_{\mathbb{P}} - \mu_{\mathbb{Q}} \rangle_{\mathcal{H}}$. A characteristic RKHS is one for which the embedding is injective: that is, each measure has a unique embedding [25], and $\|\mu_{\mathbb{P}} - \mu_{\mathbb{Q}}\|_{\mathcal{H}} = 0$ if and only if $\mathbb{P} = \mathbb{Q}$. This property holds for many commonly used kernels. For $\mathcal{T} = \mathbb{R}^d$, this includes the Gaussian kernels.

## 3 A Reinforcement Learning Framework

Suppose we are interested in modeling the daily crime patterns, or monthly occurrences of disease for patients, then the data are collected as trajectories of events within a predefined time window $T$. We regard the observed paths as actions taken by an expert (nature).

Let $\xi = \{\tau_1, \tau_2, \ldots, \tau_{N_T^\xi}\}$ represent a single trajectory of events from the expert where $N_T^\xi$ is the total number of events up to $T$, and it can be different for different sequences. Then, each trajectory $\xi \sim \pi_E$ can be seen as an expert demonstration sampled from the expert policy $\pi_E$. Hence, on a high level, given a set of expert demonstrations $\mathcal{D} = \{\xi_1, \xi_2, \ldots, \xi_j, \ldots | \xi_j \sim \pi_E\}$, we can treat fitting a temporal point process to $\mathcal{D}$ as searching for a learner policy $\pi_\theta$ which can generate another set of sequences $\tilde{\mathcal{D}} = \{\eta_1, \eta_2, \ldots, \eta_j, \ldots | \eta_j \sim \pi_\theta\}$ with similar patterns as $\mathcal{D}$. We will elaborate on this reinforcement learning framework below.

**Reinforcement Learning Formulation (RL).** Given a sequence of past events $s_t = \{t_i\}_{t_i < t}$, the stochastic policy $\pi_\theta(a|s_t)$ samples an inter-event time $a$ as its action to generate the next event time as $t_{i+1} = t_i + a$. Then, a reward $r(t_{i+1})$ is provided and the state $s_t$ will be updated to $s_t = \{t_1, \ldots, t_i, t_{i+1}\}$. Fundamentally, the policy $\pi_\theta(a|s_t)$ corresponds to the conditional probability of the next event time in temporal point process, which in turn uniquely determines the corresponding intensity function as $\lambda_\theta(t|s_{t_i}) = \frac{\pi_\theta(t - t_i|s_{t_i})}{1 - \int_{t_i}^t \pi_\theta(\tau - t_i|s_{t_i}) d\tau}$. This builds the connection between the intensity function in temporal point processes and the stochastic policy in reinforcement learning. If reward function $r(t)$ is given, the optimal policy $\pi_\theta^*$ can be directly computed via

$$\pi_\theta^* = \arg \max_{\pi_\theta \in \mathcal{G}} \ J(\pi_\theta) := \mathbb{E}_{\eta \sim \pi_\theta} \left[ \sum_{i=1}^{N_T^\eta} r(t_i) \right], \tag{2}$$

where $\mathcal{G}$ is the family of all candidate policies $\pi_\theta$, $\eta = \{t_1, \ldots, t_{N_T^\eta}\}$ is one sampled roll-out from policy $\pi_\theta$, and $N_T^\eta$ can be different for different roll-out samples.

**Inverse Reinforcement Learning (IRL).** Eq.(2) shows that when the reward function is given, the optimal policy can be determined by maximizing the expected cumulative reward. However, in our case, only the expert's sequences of events can be observed, but the real reward function is unknown. Given the expert policy $\pi_E$, IRL can help to uncover the optimal reward function $r^*(t)$ by

$$r^* = \max_{r \in \mathcal{F}} \left( \mathbb{E}_{\xi \sim \pi_E} \left[ \sum_{i=1}^{N_T^\xi} r(\tau_i) \right] - \max_{\pi_\theta \in \mathcal{G}} \mathbb{E}_{\eta \sim \pi_\theta} \left[ \sum_{i=1}^{N_T^\eta} r(t_i) \right] \right), \tag{3}$$

where $\mathcal{F}$ is the family class for reward function, $\xi = \{\tau_1, \ldots, \tau_{N_T^\xi}\}$ is one event sequence generated by the expert $\pi_E$, and $\eta = \{t_1, \ldots, t_{N_T^\eta}\}$ is one roll-out sequence from the learner $\pi_\theta$. The formulation means that a proper reward function should give the expert policy higher reward than any other learner policy in $\mathcal{G}$, and thus the learner can approach the expert performance by maximizing this reward. Denote the procedure (2) and (3) as $\text{RL}(r)$ and $\text{IRL}(\pi_E)$, accordingly. The optimal policy can be obtained by

$$\pi_\theta^* = \text{RL} \circ \text{IRL}(\pi_E). \tag{4}$$

**Overview of the Proposed Learning Framework.** Solving the optimization problem (3) is very time-consuming in that it requires to solve the inner loop RL problem repeatedly. We relieve the computational challenge by choosing the space of functions $\mathcal{F}$ for $r(t)$ to be the unit ball in RKHS $\mathcal{H}$, which allows us to obtain an analytical expression for the updated reward function $\hat{r}(t)$ given any

current learner policy $\hat{\pi}(\theta)$. This $\hat{r}(t)$ is determined by finite sample expert trajectories and finite sample roll-outs from the current learner policy, and it directly quantifies the discrepancy between the expert's policy (or intensity function) and current learner policy (or intensity function). Then by solving a simple RL problem as in (2), the learner policy can be improved to close its gap with the expert policy using a simple policy gradient type of algorithm.

# 4    Model

In this section, we present model parametrization and the analytical expression of optimal reward function.

**Policy Network.** The function class of the policy $\pi_\theta \in \mathcal{G}$ should be flexible and expressive enough to capture the potential complex point process patterns of the expert. We, therefore, adopt the recurrent neural network (RNN) with stochastic neurons [4] which is flexible to capture the nonlinear and long-range sequential dependency structure. More specifically,

$$a_i \sim \pi(a \mid \Theta(h_{i-1})), \quad h_i = \psi(Va_i + Wh_{i-1}), \quad h_0 = 0, \tag{5}$$

where the hidden state $h_i \in \mathbb{R}^d$ encodes the sequence of past events $\{t_1, \ldots, t_i\}$, $a_i \in \mathbb{R}^+$, $V \in \mathbb{R}^d$, and $W \in \mathbb{R}^{d \times d}$. Here $\psi$ is a nonlinear activation function applied element-wise, and $\Theta$ is a nonlinear mapping from $\mathbb{R}^d$ to the parameter space of the probability distribution $\pi$. For instance, one can choose $\psi(z) = \frac{e^z - e^{-z}}{e^z + e^{-z}}$ to be the tanh function, and design the output layer of $\Theta$ such that $\Theta(h_{i-1})$ is a valid parameter for a probability density function $\pi$. The output $a_i = t_i - t_{i-1}$, serves as the $i$-th inter-event time (let $t_0 = 0$), and $a_i > 0$. The choice of model $\pi$ is quite flexible, only with the constraint that the random variable should be positive since $a$ is always positive. Common distributions such as exponential and Rayleigh distributions would satisfy such constraint, leading to $\pi(a|\Theta(h_{i-1})) = \Theta(h)e^{-\Theta(h)a}$ and $\pi(a|\Theta(h_{i-1})) = \Theta(h)ae^{-\Theta(h)a^2/2}$ respectively. In this way, we specify a nonlinear and flexible dependency over the history.

The architecture of our model in (5) is shown in Figure 2. Different from traditional RNN, the outputs $a_i$ are sampled from $\pi$ rather than obtained by deterministic transformations. This is what "stochastic" policy means. Randomly sampling will allow the policy to explore the temporary space. Furthermore, the sampled time point will be fed back to the RNN. The proposed model aims to capture that the state $h_i$

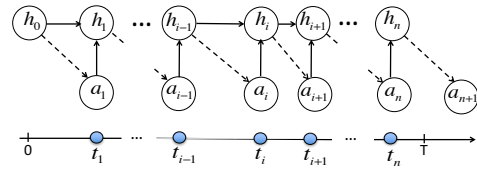

Figure 2: Illustration of generator $\pi_\theta$.

is attributed by two parts. One is the *deterministic* influence from the previous hidden state $h_{i-1}$, and the other is the *stochastic* influence from the latest sampled action $a_i$. Action $a_i$ is sampled from the previous distribution $\pi(a|\Theta(h_{i-1}))$ with parameter $\Theta(h_{i-1})$ and will be fed back to influence the current hidden state $h_i$.

In some sense, our RNN with stochastic neurons mimics the event generating mechanism of the doubly stochastic point process, such as Hawkes process and self-correcting process. For these types of point processes, the intensity is stochastic, which depends on history, and the intensity function will control the occurrence rate of the next event.

**Reward Function Class.** The reward function directly quantifies the discrepancy between $\pi_E$ and $\pi_\theta$, and it guides the learning of the optimal policy $\pi_\theta^*$. On the one hand, we want its function class $r \in \mathcal{F}$ to be sufficiently flexible so that it can represent the reward function of various shapes. On the other hand, it should be restrictive enough to be efficiently learned with finite samples [3, 13]. With these competing considerations, we choose $\mathcal{F}$ to be the unit ball in RKHS $\mathcal{H}$, $\|r\|_\mathcal{H} \leqslant 1$. An immediate benefit of this function class is that we can show the optimal policy can be directly learned via a minimization formulation given in Theorem 1 instead of the original minimax formulation (3).

A sketch of proof is provided as follows. For short notation, we denote

$$\underbrace{\phi(\eta) := \int_{[0,T)} k(t, \cdot)dN_t^{(\eta)}}_{\text{feature mapping from data space to R}}, \quad \text{and} \quad \underbrace{\mu_{\pi_\theta} := \mathbb{E}_{\eta \sim \pi_\theta}\left[\phi(\eta)\right]}_{\text{mean embeddings of the intensity function in RKHS}}$$

where $dN_t^{(\eta)}$ is the counting process associated with sample path $\eta$, and $k(t, t')$ is a universal RKHS kernel. Then using the reproducing property,

$$J(\pi_\theta) := \mathbb{E}_{\eta \sim \pi_\theta} \left[ \sum_{i=1}^{N_T^{(\eta)}} r(t_i) \right] = \mathbb{E}_{\eta \sim \pi_\theta} \left[ \int_{[0,T)} \langle r, k(t, \cdot) \rangle_{\mathcal{H}} dN_t^{(\eta)} \right] = \langle r, \mu_{\pi_\theta} \rangle_{\mathcal{H}}.$$

Similarly, we can obtain $J(\pi_E) = \langle r, \mu_{\pi_E} \rangle_{\mathcal{H}}$. From (3), $r^*$ is obtained by

$$\max_{\|r\|_{\mathcal{H}} \leq 1} \min_{\pi_\theta \in \mathcal{G}} \langle r, \mu_{\pi_E} - \mu_{\pi_\theta} \rangle_{\mathcal{H}} = \min_{\pi_\theta \in \mathcal{G}} \max_{\|r\|_{\mathcal{H}} \leq 1} \langle r, \mu_{\pi_E} - \mu_{\pi_\theta} \rangle_{\mathcal{H}} = \min_{\pi_\theta \in \mathcal{G}} \|\mu_{\pi_E} - \mu_{\pi_\theta}\|_{\mathcal{H}},$$

where the first equality is guaranteed by the minimax theorem, and

$$r^*(\cdot | \pi_E, \pi_\theta) = \frac{\mu_{\pi_E} - \mu_{\pi_\theta}}{\|\mu_{\pi_E} - \mu_{\pi_\theta}\|_{\mathcal{H}}} \propto \mu_{\pi_E} - \mu_{\pi_\theta} \tag{6}$$

can be empirically evaluated by data. In this way, we change the original minimax formulation for solving $\pi_\theta^*$ to a simple **minimization** problem, which will be more efficient and stable to solve in practice. We summarize the formulation in Theorem 1.

**Theorem 1** *Let the family of reward function be the unit ball in RKHS $\mathcal{H}$, i.e., $\|r\|_{\mathcal{H}} \leqslant 1$. Then the optimal policy obtained by (4) can also be obtained by solving*

$$\pi_\theta^* = \arg \min_{\pi_\theta \in \mathcal{G}} D(\pi_E, \pi_\theta, \mathcal{H}) \tag{7}$$

*where $D(\pi_E, \pi_\theta, \mathcal{H})$ is the maximum expected cumulative reward discrepancy between $\pi_E$ and $\pi_\theta$,*

$$D(\pi_E, \pi_\theta, \mathcal{H}) := \max_{\|r\|_{\mathcal{H}} \leqslant 1} \left( \mathbb{E}_{\xi \sim \pi_E} \left[ \sum_{i=1}^{N_T^{(\xi)}} r(\tau_i) \right] - \mathbb{E}_{\eta \sim \pi_\theta} \left[ \sum_{i=1}^{N_T^{(\eta)}} r(t_i) \right] \right). \tag{8}$$

Theorem 1 implies that we can transform the inverse reinforcement learning procedure of (4) to a simple minimization problem which minimizes the maximum expected cumulative reward discrepancy between $\pi_E$ and $\pi_\theta$. This enables us to sidestep the expensive computation of (4) caused by the solving the inner RL problem repeatedly. What's more interesting, we can derive an analytical solution to (8) given by (6).

**Finite Sample Estimation.** Given $L$ trajectories of expert point processes, and $M$ trajectories of events generated by $\pi_\theta$, mean embeddings $\mu_{\pi_E}$ and $\mu_{\pi_\theta}$ can be estimated by their respective empirical mean: $\hat{\mu}_{\pi_E} = \frac{1}{L} \sum_{l=1}^{L} \sum_{i=1}^{N_T^{(l)}} k(\tau_i^{(l)}, \cdot)$ and $\hat{\mu}_{\pi_\theta} = \frac{1}{M} \sum_{m=1}^{M} \sum_{i=1}^{N_T^{(m)}} k(t_i^{(m)}, \cdot)$. Then for any $t \in [0, T)$, the estimated optimal reward is (without normalization) is

$$\hat{r}^*(t) \propto \frac{1}{L} \sum_{l=1}^{L} \sum_{i=1}^{N_T^{(l)}} k(\tau_i^{(l)}, t) - \frac{1}{M} \sum_{m=1}^{M} \sum_{i=1}^{N_T^{(m)}} k(t_i^{(m)}, t). \tag{9}$$

Note this empirical estimator is biased at $\tau_i^{(l)}$ and $t_i^{(m)}$. Unbiased estimator can also be obtained and will be provided in **Algorithm RLPP** discussed later for simplicity.

**Kernel Choice.** The unit ball in RKHS is dense and expressive. Fundamentally, our proposed framework and theoretical results are general and can be directly applied to other types of kernels. For example, we can use the Matérn kernel, which generates spaces of differentiable functions known as the Sobolev spaces [10, 2]. In later experiments, we have used Gaussian kernel and obtained promising results.

## 5   Learning Algorithm

**Learning via Policy Gradient.** In practice, instead of minimizing $D(\pi_E, \pi_\theta, \mathcal{H})$ as in (7), we can equivalently minimize $D(\pi_E, \pi_\theta, \mathcal{H})^2$ since square is a monotonic transformation. Now, we can learn $\pi_\theta^*$ from the RL formulation (2) using policy gradient with variance reduction. First, with the likelihood ratio trick, the gradient of $\nabla_\theta D(\pi_E, \pi_\theta, \mathcal{H})^2$ can be computed as

$$\nabla_\theta D(\pi_E, \pi_\theta, \mathcal{H})^2 = \mathbb{E}_{\eta \sim \pi_\theta} \left[ \sum_{i=1}^{N_T^\eta} (\nabla_\theta \log \pi_\theta(a_i | \Theta(h_{i-1}))) \cdot \left( \sum_{i=1}^{N_T^\eta} \hat{r}^*(t_i) \right) \right], \tag{10}$$

where $\sum_{i=1}^{N_T^\eta} (\nabla_\theta \log \pi_\theta(a_i | \Theta(h_{i-1})))$ is the gradient of the log-likelihood of a roll-out sample $\eta = \{t_1, \ldots, t_{N_T^\eta}\}$ using the learner policy $\pi_\theta$.

**Algorithm RLPP**: Mini-batch Reinforcement Learning for Learning Point Processes
1. Initialize model parameters $\theta$;
2. For number of training iterations do
   - Sample minibatch of $L$ trajectories of events $\{\xi^{(1)}, \ldots, \xi^{(L)}\}$ from expert policy $\pi_E$, where $\xi^{(l)} = \{\tau_1^{(l)}, \ldots, \tau_{N_T^{(l)}}^{(l)}\}$;
   - Sample minibatch of $M$ trajectories of events $\{\eta^{(1)}, \ldots, \eta^{(M)}\}$ from learner policy $\pi_\theta$, where $\eta^{(m)} = \{t_1^{(m)}, \ldots, t_{N_T}^{(m)}\}$;
   - Estimate policy gradient $\nabla_\theta D(\pi_E, \pi_\theta, \mathcal{H})^2$ as
   $$\nabla_\theta \frac{1}{M} \sum_{m=1}^{M} \left( \sum_{i=1}^{N_T^{(m)}} \hat{r}^*(t_i^{(m)}) \log p_\theta(\eta^{(m)}) \right)$$
   where $\log p_\theta(\eta^{(m)}) = \sum_{i=1}^{N_T^\eta} (\log \pi_\theta(a_i|\Theta(h_{i-1})))$ is the log-likelihood of the sample $\eta^{(m)}$, and $r^*(t_i^{(m)})$ can be estimated by $L$ expert trajectories and $(M-1)$ roll-out samples without $\eta^{(m)}$
   $$\hat{r}^*(t^{(m)}) = \frac{1}{L} \sum_{l=1}^{L} \sum_{i=1}^{N_T^{(l)}} k(\tau_i^{(l)}, t)$$
   $$- \frac{1}{M-1} \sum_{m'=1, m' \neq m}^{M} \sum_{j=1}^{N_T^{(m')}} k(t_j^{(m')}, t);$$
   - Update policy parameters as
   $$\theta \leftarrow \theta + \alpha \nabla_\theta D(\pi_E, \pi_\theta, \mathcal{H})^2.$$

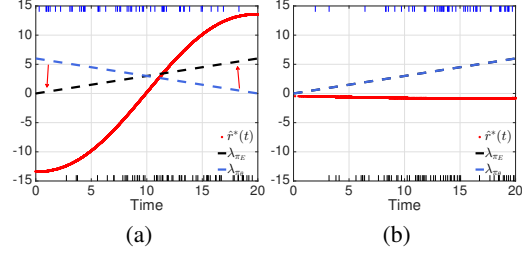

(a)  (b)

Figure 3: The reward function $\hat{r}^*(t)$ is estimated using 100 sampled sequences from $\pi_E$ and $\pi_\theta$. In (a), $\hat{r}^*(t) > 0$ when the expert's intensity is above the learner's intensity, and $\hat{r}^*(t) < 0$ when the expert's intensity is below the learner's intensity. In order to maximize the cumulative reward given the current reward, the learner should generate more events in the region when $\hat{r}^*(t) > 0$ and reduce the number of events when $\hat{r}^*(t) < 0$. Based on our formulation, the optimal reward function always quantifies the discrepancy between the expert and current learner by considering the worst case. As a result, once the learner is changed, the current optimal reward $\hat{r}^*(t)$ is updated accordingly, and $\hat{r}^*(t)$ guides the learner to update its policy towards mimicking the expert's behavior until they exactly match each other in (b) where $\hat{r}^*(t)$ becomes zero.

To reduce the variance of the gradient, we can exploit the observation that future actions do not depend on past rewards. This leads to a variance reduced gradient estimate $\nabla_\theta D(\pi_E, \pi_\theta, \mathcal{H})^2 = \mathbb{E}_{\eta \sim \pi_\theta} \left[ \sum_{i=1}^{N^\eta} (\nabla_\theta \log \pi_\theta(a_i|\Theta(h_{i-1}))) \cdot \left( \sum_{l=i}^{N_T^\eta} [\hat{r}^*(t_l) - b_l] \right) \right]$ where $\left( \sum_{l=i}^{N_T} \hat{r}^*(t_l) \right)$ is referred to as the "reward to go" and $b_l$ is the baseline to further reduce the variance. The overall procedure is given in **Algorithm RLPP**. In the algorithm, after we sample $M$ trajectories from the current policy, we use one trajectory $\eta^m$ for evaluation and the rest $M-1$ samples to estimate reward function. An example reward function learned at a different stage of the algorithm is also illustrated in Figure 3.

**Comparison with MLE.** During training, our generative model directly compares the generated temporal events with the observed events to iteratively correct the mistakes, which can effectively avoid model misspecification. Since the training only involves the policy gradient, it bypasses the intractability issue of the log-survival term in the likelihood (Eq. (1)). On the other hand, because the learned policy is in fact the conditional density of a point process, our approach still resembles the form of MLE in the RL reformulation and can thus be interpreted in a statistically principled way.

**Comparison with GAN and GAIL.** By Theorem 1, our policy is learned directly by minimizing the discrepancy between $\pi_E$ and $\pi_\theta$ which has a closed form expression. Thus, we convert the original IRL problem to a minimization problem with only one set of parameters with respect to the policy. In each training iteration with the policy gradient, we have an unbiased estimator of the gradient, and the estimated reward function also depends on the current policy $\pi_\theta$. In contrast, in GAN or GAIL formulation, they have two sets of parameters related to the generator and the discriminator. The gradient estimator is biased because each min-/max-problem is in fact nonconvex and cannot be solved in one-shot. Thus, our framework is more stable and efficient than the mini-max formulation for learning point processes.

## 6 Experiments

We evaluate our algorithm by comparing with state-of-the-arts on both synthetic and real datasets.

**Synthetic datasets.** To show the robustness to model-misspecifications of our approach, we propose the following four different point processes as the ground-truth: (I) **Inhomogeneous Poisson (IP)** with $\lambda(t) = at + b$ where $a = -0.2$ and $b = 3.5$; Here we omit $s_t$ since $\lambda(t)$ does not depend on the history. (II) **Hawkes Process (HP)** with $\lambda(t|s_t) = \mu + \alpha \sum_{t_i < t} \exp\{-(t - t_i)\}$ where $\mu = 2$, and $\alpha = 0.5$. (III) Mixture of IP and HP version 1 (**IP + HP1**). For the IP component, its $\lambda(t)$

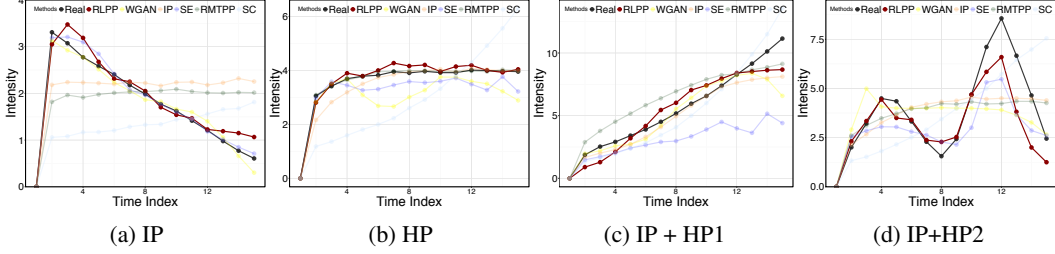

|          |          |          |          |
|----------|----------|----------|----------|
| (a) IP   | (b) HP   | (c) IP + HP1 | (d) IP+HP2 |

Figure 4: Comparison of empirical intensity functions on the synthetic data.

is piece-wise linear with monotonic increasing slopes of pieces from $\{0.2, 0.3, 0.4, 0.5\}$. The HP component has the parameter $\mu = 1$ and $\alpha = 0.5$; (IV) Mixture of IP and HP version 2 (**IP + HP2**) where the IP component also has piece-wise linear intensity but the slopes have the zig-zag pattern chosen from $\{1, -1, 2, -2\}$, and the HP component has the parameter $\mu = 1$ and $\alpha = 0.1$.

**Real datasets.** We evaluate our approach on four real datasets across a diverse range of domains:

- **911 call dataset** contains 220,000 crime incident call records from 2011 to 2017 in Atlanta area. We select one beat zone data with call timestamps ranging from 7:00 AM to 1:00 PM.
- Microsoft Academic Search (**MAS**) provides access to publication venues, time, citations, etc. We collect citation records for 50,000 papers and treat each citation time as an event.
- Medical Information Mart for Intensive Care III (**MIMIC-III**) contains de-identified clinical visit records from 2001 to 2012 for more than 40,000 patients. Our data contain 2,246 patients with at least 3 visits. For a given patient, each clinical visit will be treated as an event.
- **NYSE** contains 0.7 million high-frequency trading records from NYSE for a given stock within one day. All transactions are evenly divided into 3,200 segments. All segments have the same temporal duration. Each trading record is treated as a event.

**Baselines.** We compare our approach against two state-of-the-arts as well as conventional parametric baselines. The two state-of-the-art methods are WGANTPP [27] and RMTPP[2] [6]. In addition, three parametric methods based on maximum likelihood estimation are compared, including: (1) Inhomogeneous Poisson process where the intensity function is modeled using a mixture of Gaussian components, (2) Hawkes Process (or Self-Excitation process denoted as **SE**), and (3) Self-Correcting process (**SC**) with $\lambda(t|\boldsymbol{s}_t) = \exp\left\{\mu t - \sum_{t_i < t} \alpha\right\}$. In contrast to Hawkes process, the self-correcting process seeks to produce regular point patterns. The intuition is that while the intensity increases steadily, every time when a new event appears, it is decreased by multiplying a constant $e^{-\alpha} < 1$, so the chance of new points decreases after an event has occurred recently.

**Experimental Setup.** The policy in our method RLPP is parameterized as LSTM with 64 hidden neurons, and $\pi(a|\Theta(h))$ is chosen to be exponential distribution. Batch size is 32 (the number of sampled sequences $L$ and $M$ are 32 in Algorithm 1, and learning rate is 1e-3. We use Gaussian kernel $k(t, t') = \exp(-\|t - t'\|^2/\sigma^2)$ for the reward function. The kernel bandwidth $\sigma$ is estimated using the "median trick" based on the observations [13]. For WGANTPP and RMTPP, we are using the open source codes. For WGANTPP[3], we have used the exact experimental setup as [27], which adopts Adam optimization method [17] with learning rate 1e-4, $\beta_1 = 0.5$, $\beta_2 = 0.9$, and the batch size is 256. For RMTPP[4], batch size is 256, state size is 64, and learning rate is 1e-4.

**Comparison of Learned Empirical Intensity.** We first compare the empirical intensity of the learner point process to the expert point process. This is a straightforward comparison: one can visually assess the performance and localize the discrepancy. Fig. 4 and Fig. 5 demonstrate the empirical intensity functions of generated sequences based on synthetic and real data. It clearly shows that RLPP consistently outperforms RMTPP, and achieves comparable and sometimes even better fitting against WGANTPP. Furthermore, RLPP consistently outperforms the other three conventional parametric models when there exist model-misspecifications. Without any prior knowledge, RLPP can capture the major trends in data and can accurately learn the nonlinear dependency structure

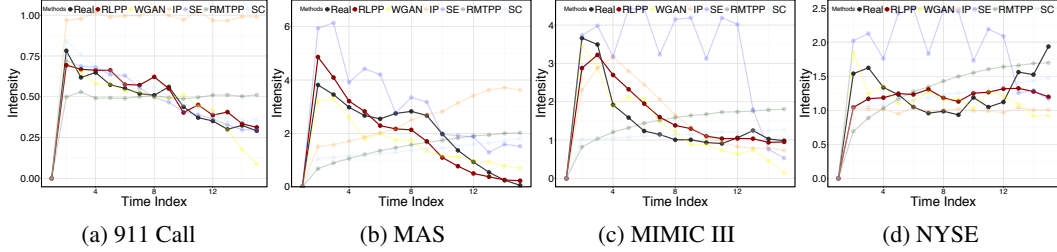

|                |                |                  |                |
|----------------|----------------|------------------|----------------|
| (a) 911 Call   | (b) MAS        | (c) MIMIC III    | (d) NYSE       |

Figure 5: Comparison of empirical intensity functions on the real datasets. For each dataset, we have used all learned models to generate new sequences. The comparisons are based on the empirical intensities estimated from the generated temporal events and those estimated from the observed temporal events.

hidden in data. In the Hawkes example, RLPP performs even as accurate as the ground-truth model. On the real-world data, the underlying true model is unknown and the point process patterns are more complicated. RLPP still shows a decent performance in the real datasets.

**Comparison of Data Fitting.** Quantile plot (QQ-plot) for residual analysis is a standard model checking approach for general point processes. Given a set of real input samples $t_1, \ldots, t_n$, by the Time Changing Theorem [5], if such set of samples is one realization of a process with the intensity $\lambda(t)$, then the respective value achieved from the integral $\Lambda = \int_{t_{i-1}}^{t_i} \lambda(t)dt$ should conform to the unit-rate exponential distribution [18]. For the synthetic experiments, since we know the exact ground-truth parametric form of $\lambda(t|s_t)$, we can perform this explicit transformation for a test. Ideally, the QQ-plot for the generated sequences should follow a 45-degree straight line. We use Hawkes Process (HP) and Inhomogeneous Poisson Process + Hawkes Process (IP+HP1) dataset to produce the QQ-plot and compare different methods in Fig. 6. In both cases, RLPP consistently stands out even without any prior knowledge about the parametric form of the true underlying generative point process and the fitting slope is very close to the diagonal line in both cases. More rigorously, we perform the KS test. Fig. 7 illustrates the cumulative distributions (CDF) of p-values. We followed the experiment setup in [21]: we generated samples from each learned point process models, transformed the time interval, and applied the KS test to compare with unit rate exponential distribution. Under this null hypothesis, the distribution of the p-values over tests should follow a uniform distribution, whose CDF should be a diagonal line. If the target distribution is the Hawkes process (Fig. 7), both the learned SE (Hawkes process) and the RLPP models are indistinguishable from that.

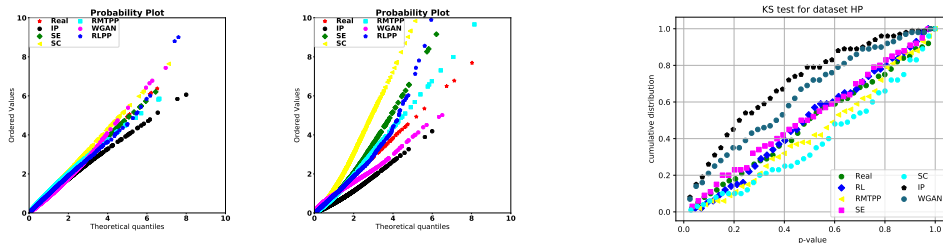

Figure 6: QQ-plot for dataset HP (left) and HP+IP1 (right).    Figure 7: KS test results: CDF of p-values.

**Comparison of Runtime.** The runtime for all methods averaged on all datasets is shown in Table 1. We note that both RLPP and WGANTPP are written in Tensorflow. However, WGANTPP adopts the adversarial training framework based on Wasserstein divergence, where both the generator and the discriminator are modeled as LSTMS. In contrast, RLPP only models the policy as a single LSTM with the reward function learned in an analytical form. As a consequence, RLPP requires less parameters and is more simpler to train while at the same time achieving comparable or even better performance.

Table 1: Comparison of runtime.

| Method | **RLPP** | WGANTPP | RMTPP | SE  | SC  | IP  |
|--------|----------|---------|-------|-----|-----|-----|
| Time   | 80m      | 1560m   | 60m   | 2m  | 2m  | 2m  |
| Ratio  | 40x      | 780x    | 30x   | 1x  | 1x  | 1x  |

**Comparisons to LGCP and non-parametric Hawkes.** We also compared RLPP to log-Gaussian Cox process (LGCP) model and non-parametric Hawkes with non-stationary background rate (Nonpar Hawkes) model regarding learned empirical intensity function. Representative comparison results are showed in Fig. 8. Our proposed method (RL) performs similarly to LGCP and outperforms Nonpar Hawkes on real datasets. However, LGCP needs to dis-

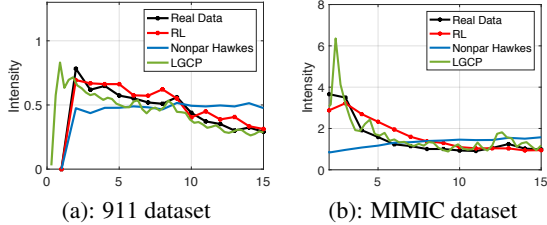

(a): 911 dataset    (b): MIMIC dataset

Figure 8: Comparison of empirical intensity functions.

cretize time into windows and aggregate event into counts. This leads to some information loss and introduces additional tuning parameters. Moreover, the standard LGCP is not scalable, typically requiring $\mathcal{O}(n^3)$ in computation and $\mathcal{O}(n^2)$ in storage ($n$ = sequence # $\times$ window #). We used an implementation in GPy package[5], which requires 50% more time than our method (127 mins *vs* 80 mins) in processing 5% of the dataset. The nonparametric Hawkes model is parametrized by weighted sum of basis functions, similar to that of the inhomogeneous Poisson process baseline, and it is difficult to generalize outside the observation window.

# 7 Discussions

1. RMTPP we compared in experiments is a state-of-the-art maximum-likelihood-based model, which uses a similar RNN outputting parametrization of exponential distributions but fits the model parameters with maximum likelihood. Across our experiments over eight synthetic and real-world datasets, our proposed method performs consistently better than the MLE.

2. In theory, although MLE has many attractive limiting properties, it has no optimum properties for finite samples, in the sense that (when evaluated on finite samples) other estimators may provide a better estimate for the true parameters, e.g. [22]. Likelihood is related to KL divergence. Since KL divergence is asymmetric and has a number of drawbacks for finite sample (such as high variance and mode dropping), many other divergences have been proposed and shown to perform better in the finite sample case, e.g. [14]. Our proposed discrepancy is inspired by a similar use of RKHS discrepancy in two sample tests in [14]. RKHS discrepancy has been shown to perform nicely on finite sample and also preserve the asymptotic properties.

3. Another potential benefit of our proposed framework is that one may use the RNN to define a transformation for the temporal random variable instead of defining its output distribution. For example, we can establish our policy as a transformation of a sample from a unit rate exponential distribution. The same empirical objective in Eq. (8) will be used, but a different optimization algorithm is needed. Since no explicit parameterization of the output distribution is needed, this may lead to even more flexible models and this is left for future investigation.

# 8 Conclusions

This paper proposes a reinforcement learning framework to learn point process models. We parametrized our policy as RNNs with stochastic neurons, which can sequentially sample discrete events. The policy is updated by directly minimizing the discrepancy between the generated sequences with the observed sequences, which can avoid model misspecification and the limitation of likelihood based approach. Furthermore, the discrepancy is explicitly evaluated in terms of the reward function in our setting. By choosing the function class of reward to be the unit ball in RKHS, we successfully derived an analytical optimal reward which maximizes the discrepancy. The optimal reward will iteratively encourage the policy to sample events as close as the observation. We show that our proposed approach performs well on both synthetic and real data.

**Acknowledgments**

This project was supported in part by NSF grants CCF-1442635, CMMI-1538746, DMS-1830210, NSF CAREER Award CCF-1650913, Atlanta Police Foundation fund, and an S.F. Express fund awarded to Yao Xie. This project was supported in part by NSF IIS-1218749, NIH BIGDATA 1R01GM108341, NSF CAREER IIS-1350983, NSF IIS-1639792 EAGER, NSF CNS-1704701, ONR N00014-15-1-2340, Intel ISTC, NVIDIA and Amazon AWS, NSF CCF-1836822, NSF IIS-1841351 EAGER, and Siemens awarded to Le Song.

## Footnotes

[2]RMTPP has very similar performance with [19].

[3]https://github.com/xiaoshuai09/Wasserstein-Learning-For-Point-Process

[4]https://github.com/dunan/NeuralPointProcess

[5] https://github.com/SheffieldML/GPy

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
