[Reviews · NeurIPS 2018]

Reviewer 1



This is a very interesting paper using inverse reinforcement learning (IRL) as a tool to learn to generate point processes (PP). The generation is very flexible and the authors argue for its merits over GAN-based and other generative methods discussed recently. Flexibly generating point processes to match the properties of data is a subject of recent interest, and this paper is an exciting contribution to it. The benefits of this method can be broadly compared to the WGAN-PP seen recently, in that the optimisation is based on distances between real and simulated PP realisations and not explicitly on likelihoods. As the authors stress, the present method seems likely to be easier to train since it is not adversarial. The authors include analytic expressions for the inner loop of their method. I am not an expert in reinforcement learning and so I mark my confidence score as lower than full. The distance function between pairs of PP realisations is absolutely crucial here, and here is specified via a choice of RKHS kernel. This choice must therefore have an effect on the quality of the results, and I would like to have seen the authors explore this, in experiments and/or in discussion. Minor feedback: * Fig 1: "are fitted back" should be "are fed back"? * line 74: the Hawkes process formula is over-simplified, constants are missing. * line 158 and 164: unexplained "f" term. * line 196: Loose wording "...the same point process" - no it is not. * line 210: "square is a monotonic transformation" - not strictly true The authors' response to reviews added further useful baseline/comparisons and dealt well with concerns raised by other reviewers.

Reviewer 2



In this paper reinforcement learning is used for non-parametric estimation of temporal point processes. The method is compared to RNN and WGAN based point processes on several synthetic and real data sets. The paper is clearly written and the method is novel. Overall from a theoretical perspective the paper is interesting. There are some problems with the baselines and experiments that can potentially be addressed. First, temporal point processes such as 911 calls are fairly low dimensional with a lot of noise. So I'm not completely convinced RNN and WGAN are the best baselines. There is a large body of statistics literature on semi and fully non parametric methods for point processes (LGCP, non-parametric Hawkes with non-stationary background rate, etc). I think these would make for better baselines as they may have better variance reduction properties and may ultimately be more accurate. The experiments also need more details and to be improved. How is cross-validation performed? In temporal point processes the split should be in time where training is done over one time period and evaluation over a subsequent time period. Is this the case here? Figure 5 is somewhat useful, but AIC values might be better. Figure 6 is hard to read, it might be better to use a KS plot.

Reviewer 3



The paper "Learning Temporal Point Processes via Reinforcement Learning" proposes a new way to lean temporal point processes, where the intensity function is defined via recurrent neural networks rather than classical parametric forms. This enables a better fitting with the true generative process. It builds on the WGAN approach but rather than dealing with a minimax optimization problem, the authors propose to use a RKHS formalization to find an analytic maximum for the expected cumulative discrepancy between processes underlying observed sequences and generated ones. The results look convincing and the followed inverse reinforcement learning approach elegant, but I am missing some justification and clarifications w.r.t. simpler models. Authors claim that their way of learning allow them to define better specified models than parametric based ones. But they still use parametric sampling of time events (e.g., using an exponential distribution for the next event). It is unclear to me how this method would compete against a baseline which would simply maximize the log-likelihood of observed sequences using a similar rnn outputing parametrizations of exponential distributions. Does this parametrization not cancel all the benefit resulting from considering the point process formalization ? For me the benefit from using Temporal point processes is that they allow one to define better specified temporal sequence distributions than using classical parametric models such as exponential laws parametrized according to the past. The problem is that they usually induce intractable likelihood functions and lead to models from which it is difficult to sample. But for me the proposed way to circumventing the problem reduces the model to classical ones... Authors clearly should add more discussion about this point. . Other remarks: - What is the baseline used to reduce the variance in the policy gradient algorithm ? A moving average of the rewards ? Authors should detail this in the paper, since they only mention a baseline without defining it afterward. - Would it be possible to consider a marked version were different kinds of events would be observed in the sequences ?